

# Simple approach for ranking structure determining residues

Oscar D. Luna-Martínez[1], Abraham Vidal-Limón[2], Miryam I. Villalba-Velázquez[1], Rosalba Sánchez-Alcalá[1], Ramón Garduño-Juárez[3], Vladimir N. Uversky[4,5,6] and Baltazar Becerril[1]

[1] Instituto de Biotecnología, Universidad Nacional Autónoma de México, Cuernavaca, Morelos, Mexico
[2] División de Biología Molecular, Instituto Potosino de Investigación Científica y Tecnológica, San Luis Potosí, Mexico
[3] Instituto de Ciencias Físicas, Universidad Nacional Autónoma de México, Cuernavaca, Morelos, Mexico
[4] Department of Molecular Medicine and USF Health Byrd Alzheimer's Research Institute, University of South Florida, Tampa, FL, United States
[5] Institute for Biological Instrumentation, Russian Academy of Sciences, Puschino, Moscow Region, Russia
[6] Laboratory of Structural Dynamics, Stability and Folding of Proteins, Russian Academy of Sciences, St. Petersburg, Russia

## ABSTRACT

Mutating residues has been a common task in order to study structural properties of the protein of interest. Here, we propose and validate a simple method that allows the identification of structural determinants; i.e., residues essential for preservation of the stability of global structure, regardless of the protein topology. This method evaluates all of the residues in a 3D structure of a given globular protein by ranking them according to their connectivity and movement restrictions without topology constraints. Our results matched up with sequence-based predictors that look up for intrinsically disordered segments, suggesting that protein disorder can also be described with the proposed methodology.

## INTRODUCTION

It is widely known that highly conserved residues in multiple sequence alignments of proteins are considered as key residues essential for protein structure and function including protein folding, phylogeny and evolutionary processes (*Phillips, Janies & Wheeler*, *2000*; *Edgar & Batzoglou*, *2006*; *Pierri, Parisi & Porcelli*, *2010*). Nevertheless, neutral mutations can accomplish great modifications to the structural properties of the protein by modifying its thermodynamic stability or biological function as seen in extremophiles in order to adapt to their environment (*Jaenicke & Böhm*, *1998*; *Rothschild & Mancinelli*, *2001*; *Reed et al.*, *2013*). Proteins can be considered as molecular entities with a tridimensional structure which are exposed to solvent and, frequently, interacting with other macromolecules. Each amino acid residue in a protein and each nucleotide base in a nucleic acid can be seen as a node and due to their high connectivity they represent hubs. Therefore, network analyses applied on protein structures attempt to assess the location of essential hubs

Corresponding author
Baltazar Becerril,
baltazar@ibt.unam.mx

that are necessary for the preservation of the stability of the global structure (*Vendruscolo et al.*, *2002*). Several mathematical approaches have been described which have helped to identify these hubs, each one employing different considerations such as geometrical approximations, thresholds, and information theory (*Costa et al.*, *2007*; *Böde et al.*, *2007*). Other approaches combine network parameters with protein properties, e.g., Relative Solvent Accesibility (RSA), to optimize the hierarchization of each residue within the structure (*Li, Wang & Wang*, *2008*). In addition, the structural relevance of each residue can be assessed by ranking the theoretical scores obtained from mathematical approximations applied over protein structures (*Greene & Higman*, *2003*; *Amitai et al.*, *2004*).

Because protein structures are not rigid molecular assemblies, X-ray structure determinations provide a "snapshot" of a "ground state," which is assumed to represent the lowest energy conformation in a crystal lattice (*Karplus & Kuriyan*, *2005*; *Rodríguez-Rodríguez et al.*, *2012*). Molecular dynamics simulations is an *in silico* tool that can provide information on detailed atomic motions at different time-scales, which have been increased through the development of more powerful hardware (*Kepleis et al.*, *2009*; *Dror et al.*, *2012*). Taking advantage of these technological advances, we sought to demonstrate that joining this important parameter with network analyses will allow the compilation of a simple method for ranking structure determining residues involved in protein stabilization. The problem of classifying each node according to its structural relevance is far from being trivial because of various reasons. For example, some residues that are not located at the hydrophobic core are known to have a long-distance effect on the structure, even in the case of a neutral mutation (*Tokuriki et al.*, *2008*; *Pace et al.*, *2011*). Our hypothesis is that the rigidness and the connectivity of each residue, irrespectively of its solvent exposure, can be associated to a specific theoretical score that can be used as a ranking parameter, where highly connected residues with restricted movement should have the greatest effects on the overall stability of globular proteins. Among all network descriptors, we selected Shannon dynamical entropy as a connectivity parameter (*Demetrius & Manke*, *2004*; *Costa et al.*, *2007*). In this study, some well-known protein structures were unbiased chosen with the goal of developing a simple method to identify, in a hierarchical way, those amino acid residues that are determinant in the structural stability of a protein. This approach basically combines molecular dynamics with a network analysis based on the Shannon dynamical entropy.

## MATERIALS AND METHODS

### Statistics

Principal component analysis was performed using the R software (*R Core Development Team*, *2013*). The thermal unfolding experiments of the 6aJL2 mutants were repeated in triplicate.

### Crystallographic structures

The PDB codes, size, and protein class of the selected structures are as follows: chymotrypsin inhibitor (2CI2; 65 residues; 16% $\alpha$ + 21% $\beta$); 6aJL2 (2W0K; chain A, 111 residues; 5% $\alpha$ + 46% $\beta$); apoflavodoxin (1FTG; 168 residues; 35% $\alpha$ + 19% $\beta$); dimeric Arc repressor (1ARR; 108 residues; 62% $\alpha$ + 9% $\beta$); DNA-binding domain of human estrogen receptor
$\alpha$ in complex with the estrogen response element DNA duplex (1HCQ; chains A, B; 128 residues; 26% $\alpha$ + 9% $\beta$, and chains C and D, 56 nucleotides); cold shock protein from *Bacillus subtilis* (1CSP; 67 residues; 4% $\alpha$ + 55% $\beta$); cold shock protein from *B. caldolyticus* (1C9O, 66 residues; 4% $\alpha$ + 62% $\beta$); and the largely disordered N-terminus of suppressor of cytokine signaling 5 mammalian suppressor of Janus Kinase interaction region (2N34, 70 residues; 12% $\alpha$). Minor structural modifications were performed on PDB files such as fulfilling N-terminal domains and incomplete lateral chains by using Swiss PDB software (*Guex & Peitsch*, *1997*). Furthermore, all nodes were renumbered in continuous order avoiding repetitions; i.e., in PDB 2CI2 the first residue begins as residue 19. Consequently, all residues were systematically renumbered so the first residue was identified as residue 1 and so forth. Solvent-exposed area was calculated with the NOC software (*Chen, Cang & Nymeyer*, *2007*).

## Grade

Residues can be regarded as nodes and contacts as edges. Hence, an edge was defined when any two non-hydrogen atoms from a pair of residues are within distance of 5 Å. To study the topology of the residue contact network, we measured the degree of node-*i*, $Ki$, as the number of neighbors of node-*i*. Chain A of crystallographic structures was selected. In the case of quaternary structures, all the structure was considered to measure the grade of each node but only chain A was selected for further comparisons.

## Molecular dynamics

The query protein was prepared using the Protein Preparation Wizard in Maestro 9.2 package (Schrödinger LCC, NY, USA) and included in a 10-Å water box that contained 14513 SPC-type water molecules for 2CI2, 16667 for 6aJL2, 20051 for 1FTG, 20990 for 1ARR, 28142 for 1HCQ, 12219 for 1CSP, and 12033 for 1C9O. Neutralizing ions were added, and other metal ions already present in the protein structure were left at the same place. The simulated annealing calculations and data analysis were conducted using the Desmond and Maestro programs, respectively (Maestro-Desmond Interoperability Tools, version 3.0; Schrödinger, NY, USA). The OPLS_2005 force field was used for every molecular dynamics simulation. Cubic periodic boundary conditions were used for most of the proteins (53.9 × 53.9 × 53.9 Å³ for 2CI2, 56.8 × 56.8 × 56.8 Å³ for 6aJL2, 60.9 × 60.9 × 60.9 Å³ for 1FTG, 61.2 × 61.2.9 × 61.2 Å³ for 1ARR, 51.1 × 51.1 × 51.1 Å³ for 1CSP, and 50.8 × 50.8 × 50.8 Å³ for 1C9O); due to the size of the complex 1HCQ, rectangular cuboid boundary conditions were used (58.5 × 59 × 91 Å³ for 1HCQ). Each simulation was adjusted with an NPT ensemble by weak coupling to an external bath temperature at constant pressure of 1 atm and relaxation time of 2 ps, regulated by Berendsen barostat (*Berendsen et al.*, *1984*). All short-range interactions were computed using a 9 Å cutoff, and for long-range interactions (electrostatic and Van der Waals), a smooth particle mesh Ewald method with a tolerance of $1 \times 10^{-9}$ was applied (*Essmann et al.*, *1995*). To ensure that our simulations started from local minima, a simulated annealing algorithm was performed. This method started the simulation at high temperature (400 K) to overcome thermodynamic and conformational barriers, followed by gradual

cooling (annealing) to reach low energy regimes. It is widely used for the optimization of structures from experimental methods, comparative protein modeling, or studying the conformational dynamics of protein or peptide folding and unfolding (*Mori & Okamoto*, *2009*). The full system was heated at 10 K for 30 ps, 100 K for 100 ps, 300 K for 200 ps, 400 K for 300 ps, and 400 K for 500 ps and then cooled to 298 K for 1,000 ps. To ensure that heating at 400 K did not affect protein or protein/DNA structure, the RMSD of heavy atoms (C, N, O, S, P) derived from the annealed structure was compared against corresponding crystallographic structure. If RMSD standard deviation was ≤1.0 Å, then it was assumed that the annealing algorithm did not changed the whole structure or denature it. In fact, since the displacement of heavy atoms above 1 Å is considered as a conformational change, the algorithm can be trusted in finding the local minimum. A lineal interpolation step between two adjacent time points was employed. After the sixth step, a production of 25 ns was achieved with an integration time of 1 fs. The entire analysis was performed using trajectory coordinates, and the energies were written to a disk every 1.2 ps. A frame was extracted every 0.25 ns throughout the simulation, and the overall frames were saved as a PDB file.

## Dynamical entropy

An analytical algorithm encoded in the Perl language was developed to generate three files using the 100-frames PDB-file as the template. In the first step, a single file is generated for each model, and this single file contains the atom coordinates, the corresponding molecular weight, and the name of the node (amino acid or nucleotide base). The second step calculates the coordinates of the center of mass of the node. The third step calculates the distance between the center of mass of each pair of nodes and their normalized distance described by Eq. (1),

$$\overline{d_{ij}} = \frac{d_{ij}}{r_{vdW_i} + r_{vdW_j}} \tag{1}$$

where $\overline{d_{ij}}$ is the normalized distance, $d_{ij}$ is the distance between the center of mass of node $i$ and that of node $j$, and $r_{vdW}$ is the Van der Waals radius of the respective node considering all their atoms. For amino acid residues, values of their Van der Waals radiuses comprehending the whole residue were obtained from *Darby & Creighton* (*1993*). For nucleotide bases, values were obtained from *Voss & Gerstein* (*2005*). The mean distance between each pair of nodes of the 100 structures sampled was calculated. Since the next steps involve eigenvector and eigenvalue measurement properties, the mean value of the inverse normalized distance was calculated, so the largest weight represents the closest distance between a pair of nodes and the smallest weight represents the longest distance.

A weighted adjacency matrix $A = (a_{ij}) \geq 0$ of size $N \times N$, where $N$ is the number of nodes, was constructed. In this case, matrix $A$ is symmetric ($a_{ij} = a_{ji}$) and undirected. Following the mathematical strategy described in *Demetrius & Manke* (*2004*) we now assume that the stochastic process is given by a Markov Matrix $P = p_{ij}$ where $p_{ij} \geq 0$ and $\Sigma_j p_{ij} = 1$. The stationary distribution of matrix $P$ is described by Eq. (2),

$$\pi P = \pi \tag{2}$$

where $\pi$ is defined as the left-hand eigenvector associated with the largest eigenvalue solved with Mathcad 15$^{\text{TM}}$ software. The dynamical entropy of this process of each node, $H_i$, is described by Eq. (3),

$$H_i = -\sum_j \pi_i p_{ij} \log p_{ij} \tag{3}$$

where the term $p_{ij} \log p_{ij}$ is the standard Shannon entropy.

## 6aJL2 mutants

The synthesis of single-point mutants of 6aJL (Arg25His, Ile30Gly, Tyr36Phe, and Gln6Asn) was performed using recursive PCR (*Prodromou & Pearl*, *1992*). The obtained DNAs were cloned into the pSyn1 expression vector (*Schier et al.*, *1995*). All of the constructions were verified by nucleotide sequencing (*Sanger, Nicklen & Coulson*, *1977*). The variants were expressed in *Escherichia coli* BL21 (DE3) and purified as described previously (*Del Pozo-Yauner et al.*, *2008*). The protein purity was verified using SDS-PAGE electrophoresis, and the protein concentration was determined spectrophotometrically at 280 nm in 6.5 M GdnHCl and 20 mM sodium phosphate buffer, pH 7.5, using molar extinction coefficients calculated from the amino acid sequence using the ProtParam software, which is available at the ExPASy website (*Artimo et al. al.*, *2012*).

## Unfolding

Samples containing 50 µg/ml of protein in phosphate-buffered saline (PBS), pH 7.5, were placed into a 3-ml quartz cuvette. Changes in the tryptophan fluorescence were measured using a LS50B Perkin Elmer spectrofluorometer with an excitation wavelength of 295 nm (2.5-mm bandwidth) and an emission wavelength of 355 nm (5-mm bandwidth). The temperature was increased from 298 to 350 K at a rate of 1 K/min and then samples were cooled to 298 K at the same rate. The data were analyzed using the thermal unfolding Eq. (4), which was obtained from *Eftink* (*1995*).

$$F_{Trp} = \frac{(y_n + m_n T) + (y_d + m_d T) e^{\left(\frac{\Delta H_m}{RT_m} - \frac{\Delta H_m}{RT}\right)}}{1 + e^{\left(\frac{\Delta H_m}{RT_m} - \frac{\Delta H_m}{RT}\right)}} \tag{4}$$

where $F_{Trp}$ is the tryptophan fluorescence, $T$ is the temperature, $T_m$ is the temperature of the midpoint, $\Delta H_m$ is the enthalpy at $T_m$, $y_n$ and $m_n$ describe the pre-transition phase, and $y_d$ and $m_d$ describe the post-transition phase. Non-linear regression was performed using the OriginPro8$^{\text{TM}}$ software. The change in the Gibbs energy of the wild-type versus the mutant ($\Delta\Delta G$) was calculated for temperature- and denaturant-induced unfolding processes using the following equations.

Thermal unfolding (*Becktel & Schellman*, *1987*), Eq. (5):

$$\Delta\Delta G = \frac{\Delta H_{m\text{WT}}}{T_{m\text{WT}}} (T_{m\text{MUT}} - T_{m\text{WT}}) \tag{5}$$

where WT refers to the wild-type values and MUT refers to the mutant values of the melting temperature $T_m$, and $\Delta H_m$ is the enthalpy value.

Chemical unfolding (*Creighton*, *1990*), Eq. (6):

$$\Delta\Delta G = m_{\text{WT}}(C_{m\text{MUT}} - C_{m\text{WT}}) \tag{6}$$

where $m_{WT}$ is the transition slope of the wild-type, and $C_m$ is the denaturant concentration at which $\Delta G = 0$.

## RESULTS AND DISCUSSION

Many of the residues located at the hydrophobic core are considered essential (deleterious if mutated), but it is hard to identify which one will be lethal through simple visual inspection of a protein structure. In most cases reported so far, mutations increase the amount of unstable conformers, rendering a protein more susceptible to external perturbations; and this is more likely if the mutations are located at the inner core of the protein (*Jackson et al.*, *1993*; *Baldwin & Matthews*, *1994*; *Lei & Duan*, *2004*; *Kumar & Nussinov*, *2001*; *Reed et al.*, *2013*). Additionally, evaluating the role of each position by replacing the corresponding residue with any other amino acid becomes a more challenging task as the protein becomes bigger. Therefore, we sought to generate a simple method to assess proteins independently of their structural complexity (secondary, tertiary or even quaternary structure).

We selected several proteins with diverse topologies that have been thermodynamically characterized to evaluate the effects of the incorporation of some single-point mutations near neutral pH. Chymotrypsin inhibitor, 6aJL2, apoflavodoxin, arc repressor, estrogen receptor/DNA estrogen/response element complex (ECR), cold shock protein of *B. caldolyticus* (Csp C), cold shock protein of *B. subtillis* (Csp S), and the largely disordered N-terminus domain of Janus Kinase interaction region were the representative proteins chosen for this study (see Methods section for PDB details). Not only these proteins were selected because they exhibit different folds, but also because they were analyzed using distinct experimental procedures that provided different thermodynamic parameters covering two-state and three state unfolding pathways. Most of the selected proteins have been thermodynamically evaluated elsewhere through denaturation experiments using single-point mutants (*Milla, Brown & Sauer*, *1994*; *Itzhaki, Otzen & Fersht*, *1995*; *Perl & Schmid*, *2001*; *Banci et al.*, *2004*; *Campos et al.*, *2004a*; *Campos et al.*, *2004b*; *Wunderlich, Martin & Schmid*, *2005*; *Wunderlich & Schmid*, *2006*; *Gribenko & Makhatadze*, *2007*; *Del Pozo-Yauner et al.*, *2008*; *Deegan et al.*, *2010*; *Hernández-Santoyo et al.*, *2010*; *Van den Bedem*, *2013*; *Del Pozo-Yauner et al.*, *2014*). $\Delta\Delta G$ value represents the effect of a determined mutation on the stability compared to the wild-type protein structure. On the other hand, the estrogen receptor complex (ERC) was assessed by determining the affinity constant ($K_D$) against its target DNA. The percentage difference in terms of $K_D$ determines the variation level in affinity between the wild-type ER homodimer and its mutant homodimer when binding to its target DNA sequence. Both, percentages lower than 100 in $K_D$ and negative $\Delta\Delta G$ values indicate less stable structures.

We compared these experimental data with a network parameter- Grade (see File S1). This parameter is defined as the number of neighbors of each node when any two non-hydrogen atoms from residues *i* and *j* are within a cutoff distance (*Li, Wang & Wang,*

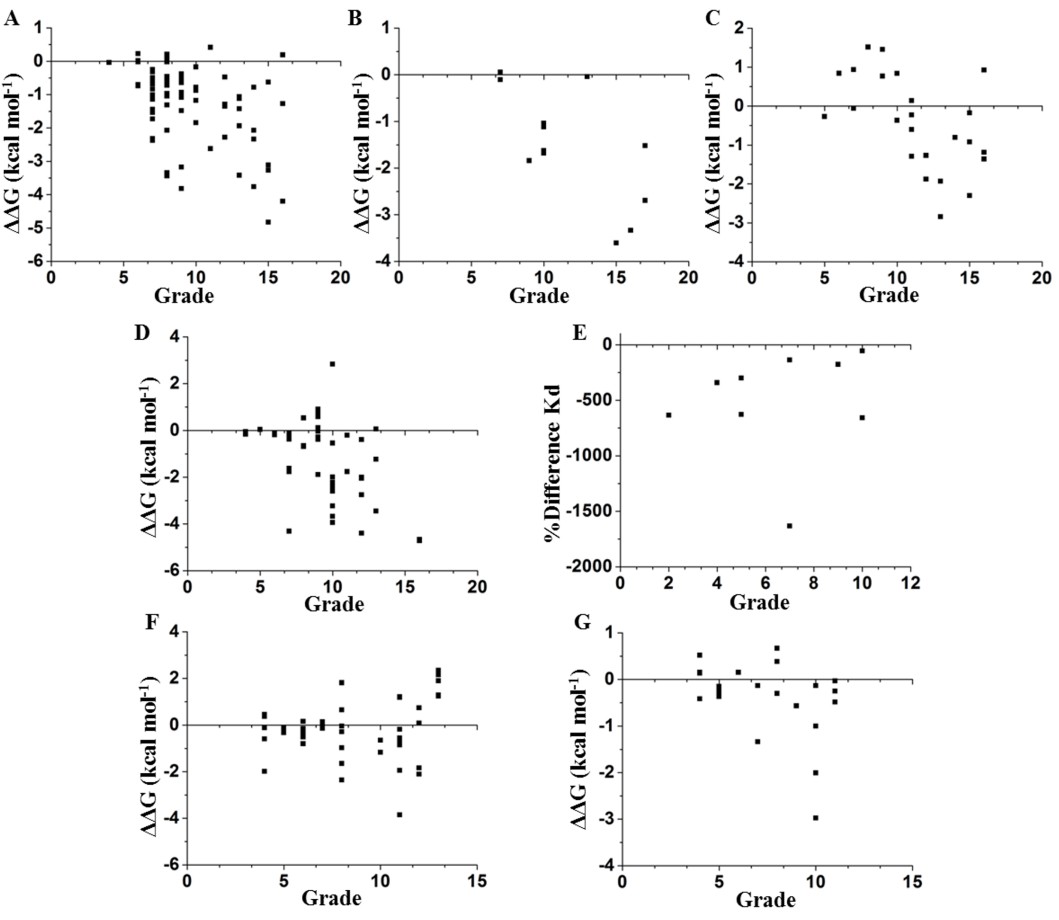

**Figure 1 Contrasting of network parameter grade against experimental values.** Each point in the graphics represents scores determined from the number of contacts surrounding the residue where the mutation was performed (see File S1). (A) Chymotrypsin inhibitor; (B) 6aJL2; (C) apoflavodoxin; (D) arc repressor; (E) DNA-binding domain of the estrogen receptor *w*; (F) cold shock protein from *B. subtilis*; and (G) cold shock protein from *B. caldolyticus*.

*2008*). According to Fig. 1, there is a higher probability to affect the stability if the selected node is highly surrounded in a cutoff distance of 5 Å. However, this parameter does not provide a full hierarchization of each node; some nodes can share same grade-value but exhibit a different impact on the stability. Another caveat is that size matters, because bigger residues, such as tryptophan or phenylalanine, tend to make more interactions than smaller residues, such as glycine or alanine (see File S1). Furthermore, proteins are not rigid structural assemblies since most of their contacts are in a dynamical condition. Thus, we suggest that the molecular dynamics of a protein should be gathered with more precise network analyses in order to properly assess the influence of determined mutations on the stability of the protein structure.

As a first step to assess this assumption, the crystallographic structures were subjected to 25 ns of molecular dynamics simulations as described in the Methods section. The completion of this process was confirmed by inspecting the Root Mean Square Deviation (RMSD) values of the main chain indicating the convergence to a stationary structural

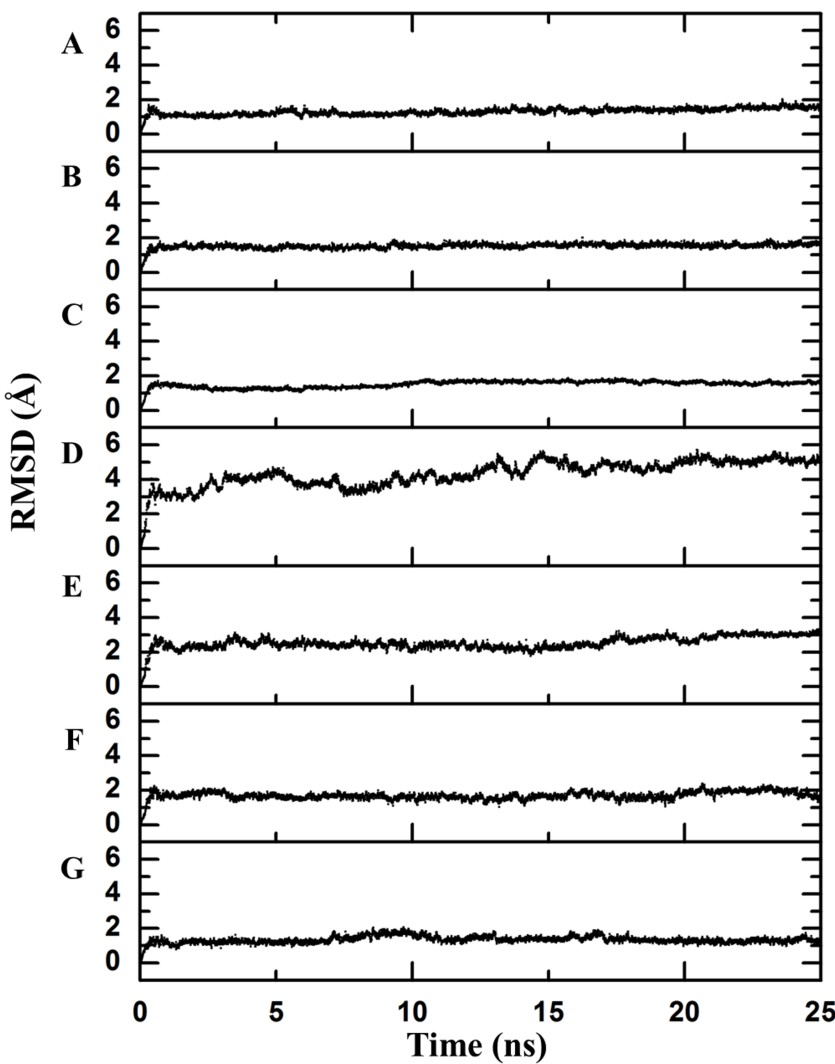

**Figure 2** **Backbone RMSD.** RMSD difference between the backbone of the crystallographic structure and the corresponding structure present at the indicated time of molecular dynamics; this difference was calculated using Desmond. There is an observable change after the first few nanoseconds of the dynamics simulations, during which the structure is "heated," but the protein remains stable after that moment. (A) Chymotrypsin inhibitor; (B) 6aJL2; (C) apoflavodoxin; (D) arc repressor; (E) DNA-binding domain of the estrogen receptor; (F) cold shock protein from *B. subtilis*; and (G) cold shock protein from *B. caldolyticus*.

movement at 300 K (Fig. 2). Every 0.25 ns, a structure was extracted by means of generating one hundred frames over all the simulation time in order to obtain a good representation of the possible movements of both the lateral and the main chains. Next, a network analysis was performed for each frame. In accordance with definitions, each protein residue and each nucleotide base were considered as individual nodes. The center of mass was calculated based on all of the atoms of the node. The interaction strength between a pair of nodes was measured in terms of the distance between their mass centers which was subsequently normalized by the summation of their Van der Waals radii considering all their atoms, as

**Table 1  Statistics of theoretical scores—SDRIs.** Skewness, kurtosis, and the normality of the SDRI distribution measured with Shapiro–Wilk test. SDRI distribution of globular proteins shows a good distribution unlike unstructured peptide. Analysis was performed using SigmaPlot11.0 software (Systat Software, San Jose, CA).

| Protein | Skewness | Kurtosis | Shapiro–Wilk |
|---|---|---|---|
| Chymotrypsin inhibitor | 0.16 | −0.59 | 0.98 |
| 6aJL2 | 0.74 | −0.03 | 0.95 |
| Apoflavodoxin | 0.80 | 0.12 | 0.95 |
| Arc repressor | 0.51 | 0.02 | 0.97 |
| Complex estrogen/Receptor | −0.11 | −0.19 | 0.99 |
| Cold shock protein BS | 0.53 | −0.67 | 0.94 |
| Cold shock protein BC | 0.61 | −0.54 | 0.94 |
| Unstructured peptide | 1.75 | 2.15 | 0.71 |

stated in Materials and Methods section (values less than 1 imply that the interaction is very strong).

We selected dynamical Shannon entropy as an approach to estimate connectivity being aware that the results describe probabilistic values. The next step in the calculation process included the use of eigenvector properties for which, the greater value, the more important interaction. The inverse value of the mean normalized distance for the 100 frames was calculated and rounded up to four decimals. At this point, we would like to emphasize that this strategy makes a cutoff distance unnecessary. Then, each of these values was incorporated into a square matrix that was, in turn, converted into a row stochastic matrix. Assuming that the microscopic process of the network is Markovian, the matrix was solved consistently using a dynamical entropy equation corresponding to a Markov process (*Demetrius & Manke*, *2004*; *Costa et al.*, *2007*). Each node is now associated with a dynamical entropic value ($H_i$), which can be interpreted as a connectivity parameter with a probabilistic character. We also presume that the movement restriction of each node, which is represented by its Root Mean Square Fluctuation (RMSF) value derived from the molecular dynamics, is associated with its dynamical entropic value. Therefore, each node was scored by dividing its entropic value by its respective RMSF value, and this score indicates the relative importance of each node within its respective structure (see File S2). These theoretical scores were identified as structural-determining residue identifier (SDRI). By normalizing SDRIs and plotting them against their sequence, SDRIs were distributed throughout the structure (Fig. 3). Distribution statistics- skewness, kurtosis, and Shapiro–Wilk test showed that globular proteins are near normal distribution unlike the unstructured peptide which is described hereinafter (Table 1). Interestingly, structures in complex—Arc repressor and Complex Receptor DNA, showed fewer residues with low SDRI values compared to the other analyzed structures, suggesting that both are notably less disordered.

We compared the distribution of the non-normalized SDRIs obtained in our study with scores from the structure-based flexibility as well as the sequence-based intrinsic disorder predisposition of the query proteins. The structural flexibility of these proteins was obtained

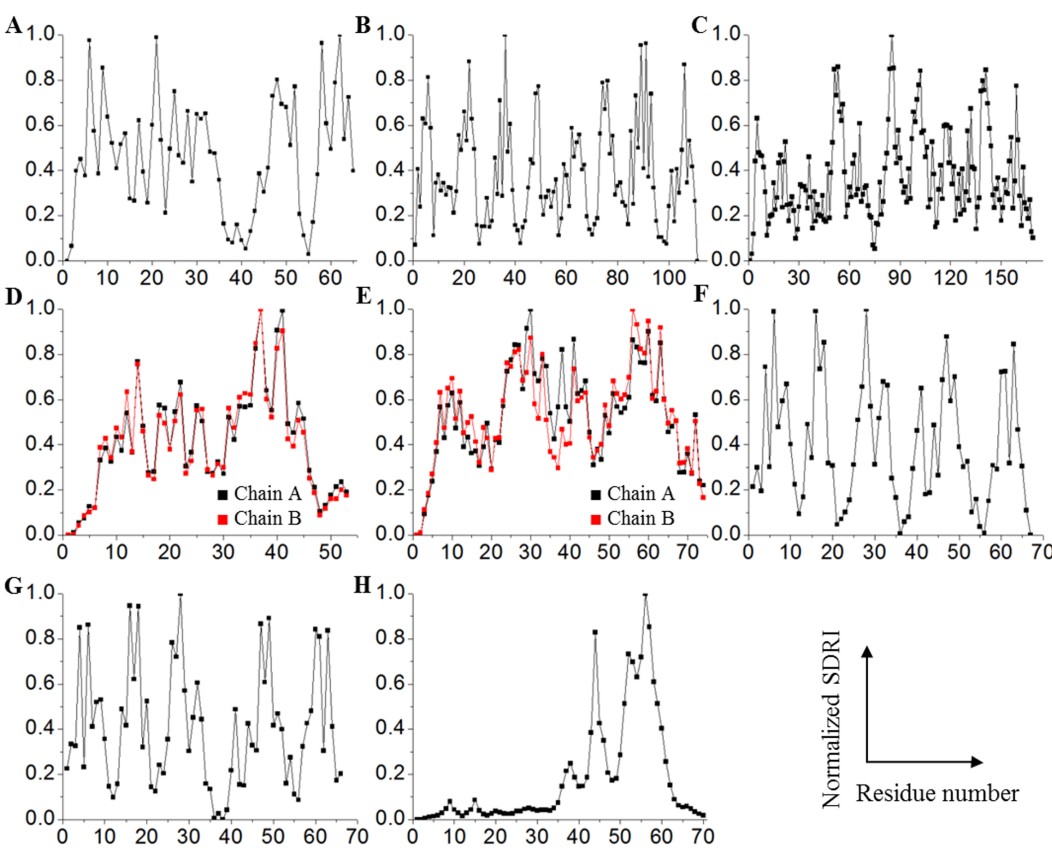

**Figure 3  Contrasting of SDRIs against their corresponding sequence.** Each point in the graphics represents normalized SDRIs from the theoretical analysis according to their sequence position (see File S2). Values near 0 indicate flexible and lesser connected residues while values near 1 indicate rigid and well-connected residues: (A) Chymotrypsin inhibitor; (B) 6aJL2; (C) apoflavodoxin; (D) arc repressor; (E) complex of estrogen receptor α/DNA estrogen response element; (F) cold shock protein from *B. subtilis;* (G) cold shock protein from *B. caldolyticus*; and (H) the JAK interaction region of SOCS5.

by utilizing the FlexPred tool that predicts the absolute fluctuations per-residue from a three-dimensional structure using the B–factors of a query protein (*Jamroz, Kolinski & Kihara*, 2012). The intrinsic disorder propensities per-residue of these proteins was obtained by using PONDR® VSL2B predictor, which is one of the more accurate standalone disorder predictors (*Fan & Kurgan*, 2014; *Peng et al.*, 2005; *Peng & Kurgan*, 2012). Results of these comparisons are shown in Fig. 4 and clearly illustrated that these three computational tools, SDRI, FlexPred, and PONDR® VSL2B, can "see" different, although related, features in a protein. Note that we used (1–SDRI) function instead of SDRI when representing the SDRI values to compare data from these three tools "in phase" to highlight intrinsic disorder instead of structural rigidness. These results suggested that there is a good agreement between the structural flexibility calculated from the protein crystal structure and the propensity of a protein to preserve disorder. Furthermore, it seemed that residues essential for the preservation of the stability of global protein structure are typically located within highly ordered and less flexible domains. In terms of spectral analysis, the visual inspection of plots shown in Fig. 4 suggested that, in many cases, the propensity for intrinsic

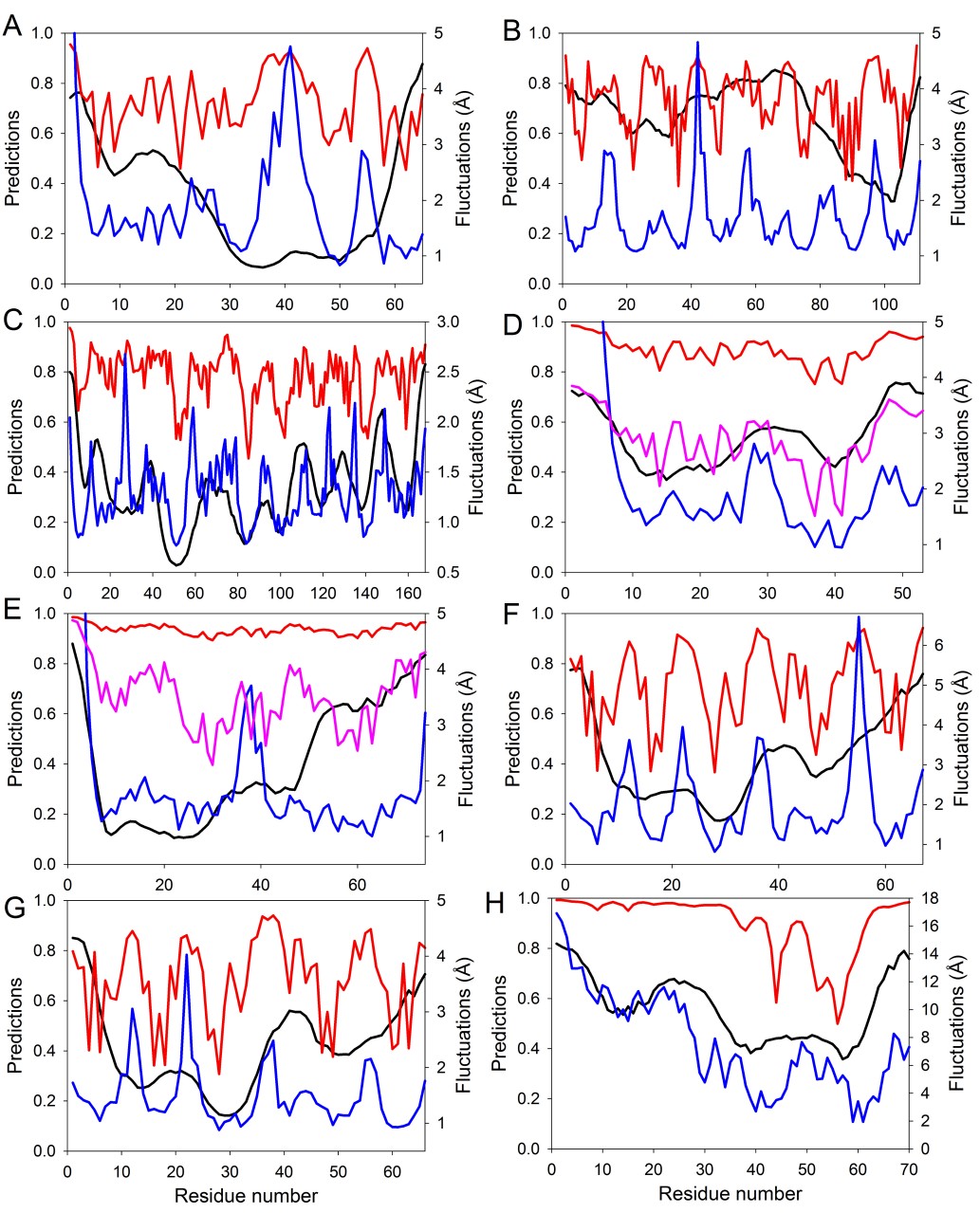

**Figure 4** **Evaluating intrinsic disorder and structural flexibility.** Distributions of predicted intrinsic disorder propensity evaluated by PONDR® VSL2B (black curves), predicted structural flexibility estimated by FlexPred (blue curves) and (1-SDRI) function ranking structure determining residues (red or pink curves) for a set of query proteins: (A) Chymotrypsin inhibitor; (B) 6aJL2; (C) apoflavodoxin; (D) arc repressor; (E) DNA-binding domain of the estrogen receptor α; (F) cold shock protein from *B. subtilis*; (G) cold shock protein from *B. caldolyticus*; and (H) the JAK interaction region of SOCS5. Propensities for intrinsic disorder and (1-SDRI) function are scaled from 0 to 1. Since formation of two complexes— dimeric Arc repressor and a complex between the estrogen receptor DNA-binding domain and the DNA estrogen response element—resulted in a dramatic reduction of the amplitude of the (1-SDRI) function, corresponding plots (D and E) also include expanded (1- SDRI) curves shown in pink.

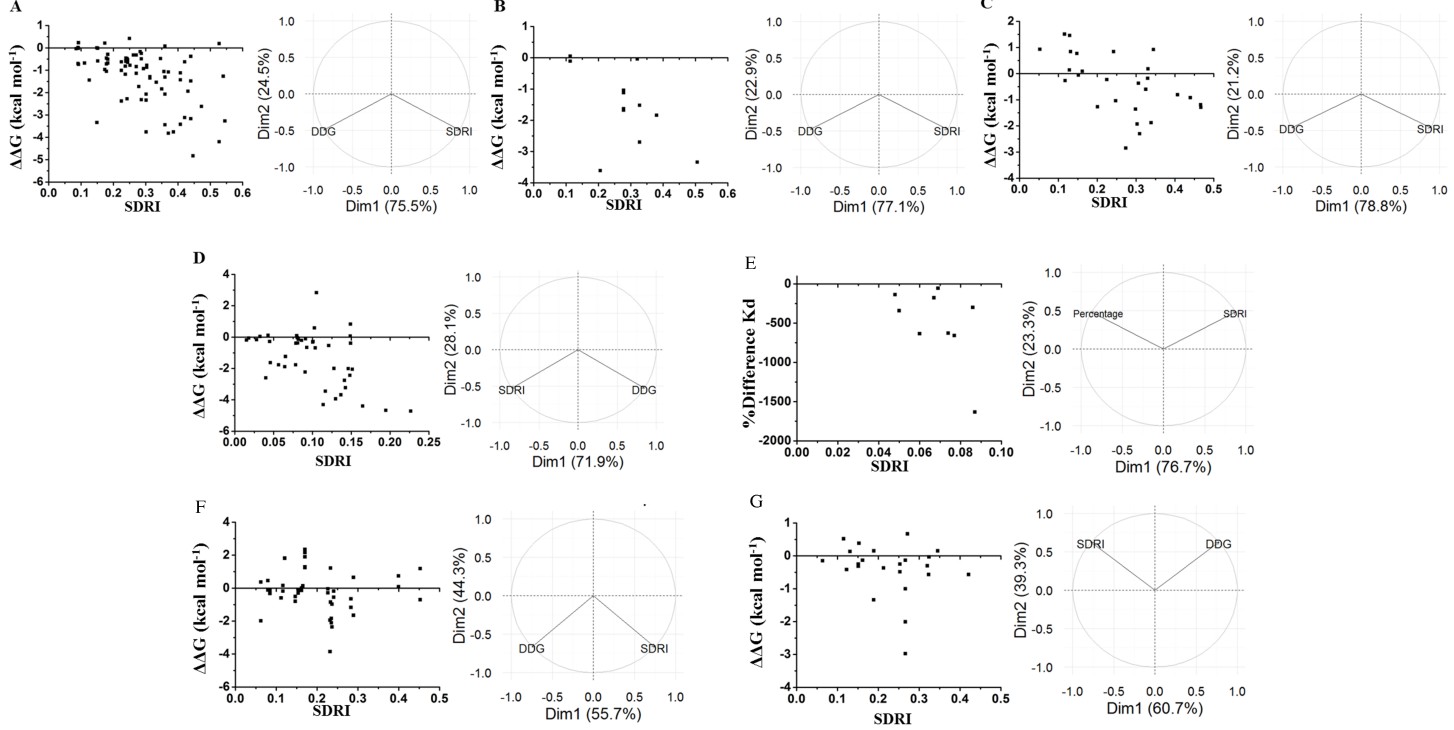

**Figure 5  Contrasting of SDRIs against their corresponding experimental values.** Each point in the graphics represents a single-mutation experiment compared with the scores from the theoretical analysis (see File S3). In the case of homodimers, only the values for chain A are shown. Statistical data is shown in terms of eigenvectors: DIM1 refers to SDRIs ($H_{i*}\mathrm{RMSF}_i^{-1}$), and DIM2 refers to the experimental values, $\Delta\Delta G$ (DDG) or KD percentage difference. (A) Chymotrypsin inhibitor; (B) 6aJL2; (C) apoflavodoxin; (D) arc repressor; (E) complex of estrogen receptor $\alpha$/DNA estrogen response element; (F) cold shock protein from *B. subtilis;* and (G) cold shock protein from *B. caldolyticus.*

disorder results in broad bands that define global appearance of the curves, whereas the outputs of (1-SDRI) and FlexPred add fine structural resolution to the resulting plots. Additionally, for some regions, noticeable disagreements can be found among the outputs of these three tools, which can be attributed to the particular considerations of each tool. Nevertheless, these important observations suggested that intrinsic disorder propensity, predicted from amino acid sequence, serves as an important background defining global flexibility of a protein 3D-structure which is fine-tuned by long-distance interactions taking place in a folded molecule.

A more efficient way to exploit SDRIs was to visualize each node according to its theoretical score; the higher SDRI the more important is the residue to preserve the structure. In homodimeric domains (arc repressor and ERC) both monomers showed the same distribution. In our analysis, the residues with highest SDRI values corresponded to structure-determinant residues (Fig. 5). Complementarily, heat maps based on these values facilitated the localization of essential residues or segments involved in a protein's biological function (Fig. 6). Despite that linear statistics parameters (R-squared and Pearson Correlation Coefficient) showed low correlation between SDRIs and experimental values (Table 2), visual inspection of Fig. 5 strongly suggested that a pattern was followed. Therefore, we performed a statistical scrutiny applying Principal Component Analyses

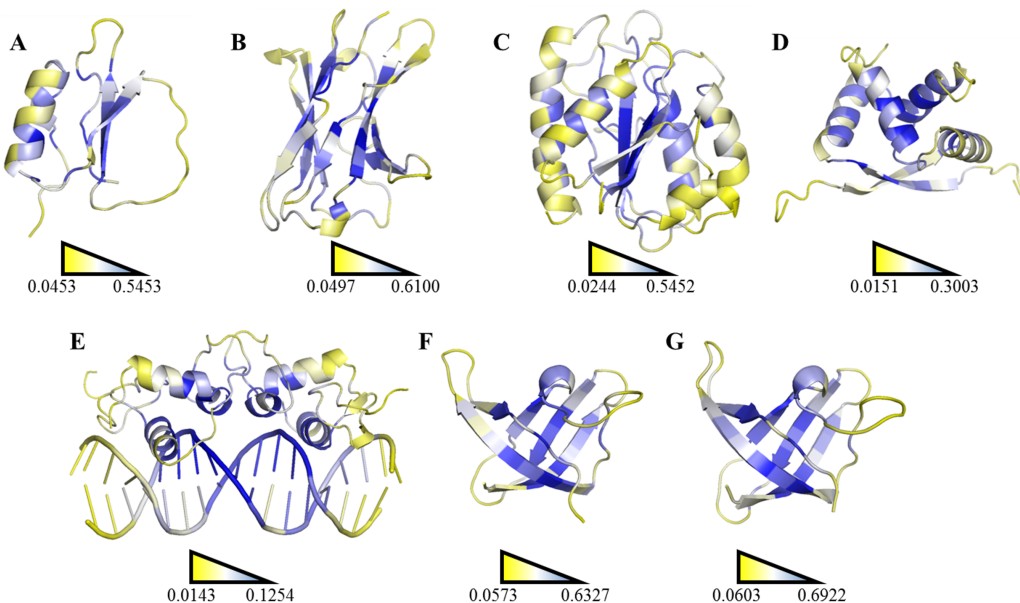

**Figure 6** **Ranking heat map.** The highest SDRI, representing a highly connected node with restricted movement, is shown in blue, and the lowest theoretically scored node, which is barely connected and exhibits a high degree of movement, is represented in yellow (see File S2). The structures were prepared using the PyMOL software. (A) Chymotrypsin inhibitor (PDB entry 2CI2); (B) 6aJL2 (PDB entry 2W0K); (C) apoflavodoxin (PDB entry 1FTG); (D) arc repressor (PDB entry 1ARR); (E) complex of estrogen receptor $\alpha$/DNA estrogen response element (PDB entry 1HCQ); (F) cold shock protein from *B. subtilis*, and (G) cold shock protein from *B. caldolyticus*.

**Table 2** **Statistics of SDRIs versus experimental results.** Coefficient of determination ($R^2$) and Pearson correlation coefficient analysis describe mild correlation between SDRIs and experimental values. Analysis was performed using SigmaPlot11.0 software (Systat Software, San Jose, CA).

| Protein | $R^2$ | Pearson correlation coefficient |
|---|---|---|
| Chymotrypsin inhibitor | 0.25 | −0.50 |
| 6aJL2 | 0.22 | −0.54 |
| Apoflavodoxin | 0.28 | −0.55 |
| Arc repressor | 0.24 | −0.50 |
| Complex estrogen/Receptor | 0.29 | −0.54 |
| Cold shock protein BS | 0.06 | −0.12 |
| Cold shock protein BC | 0.08 | −0.21 |

(PCA) to identify data patterns of apparently uncorrelated variables (*Abdi & Williams, 2010*). Results were represented as circular biplots allowing us to examine the correlation between variables SDRI and experimental data as vectors (see Fig. 5 and Fig. S2). Based on biplots from PCA results and due to the angle between variables SDRIs and thermodynamic data, it can be emphasized that the present method is able to estimate the effect of a single-point mutation on protein structure depending on the importance of a given residue irrespective of its position. The results confirmed our hypothesis that the most connected and the most rigid residues are the most influential on the structural stability of the protein

despite their involvement in any kind of structural organization. For example, the worst effects in Arc repressor were observed in five mutants (VA22, EA36, IA37, VA41, and FA45) that showed little or no cooperation in denaturation experiments (*Milla, Brown & Sauer*, *1994*). Remarkably, these mutations were performed on residues that matched with high SDRI values (6, 4, 1, 2, and 19, respectively).

The next comparison was between the SDRIs of two structures that have different thermodynamic stability in spite of their high sequence and high structural homology. We selected the Cold shock protein from thermophilic *B. subtilis* (Csp S) and from hyperthermophilic *B. caldolyticus* (Csp C) bacteria. Csp C and Csp S share a sequence homology >80%, but the hyperthermophilic variant, Csp C, is more stable than its thermophilic counterpart. Perl and Schmidt generated mutants in Csp S by directing them to Csp C sequence (see Fig. S1) (*Perl & Schmid*, *2001*). Our results show that Csp S stabilizing mutations were performed over low connected and highly flexible residues, residues with low SDRIs. Most stabilizing mutations of Csp S, that were directed to the sequence of the hyperthermophilic variant- Csp C, were those incorporated on the surface bonding flexible residues through the formation of salt bridges. Likewise, Tokuriki and Tawfik reported that mutations on surface residues of their analyzed proteins resulted in low destabilizing effects while mutations on core residues caused stronger destabilizing effects (*Tokuriki & Tawfik*, *2009*).

We are particularly interested in the characterization of 6aJL2, an immunoglobulin light chain variable domain, based on the fact that *6a* is the most implicated germ line in AL amyloidosis disease (*Comenzo et al.*, *2001*). Destabilizing mutations of 6aJL2 enhances its propensity to generate protein fibers. Strikingly, the crystallographic structures of destabilizing mutants exhibited a low RMSD difference when overlapped against the wild-type structure (*Hernández-Santoyo et al.*, *2010*). To experimentally demonstrate our hypothesis, we selected four residues with low RSA values but different SDRIs to perform single-point mutations: Gln6Asn, Arg24His, Tyr36Phe, and Ile29Gly (SDRIs 0.5055, 0.3273, 0.3274, and 0.2057, respectively). Despite that the greatest destabilization impact was detected when the size of the lateral chain was minimized, as observed with mutant Ile29Gly (RSA 2.2%, $\Delta\Delta G = -3.61$ kcal mol$^{-1}$), mutant Gln6Asn (RSA 12%, $\Delta\Delta G = -3.34$ kcal mol$^{-1}$) was more relevant due to the neutrality of the change. The lateral chain size-reduction by one methylene had a remarkable impact on the protein stability. We selected residue Gln6 in 6aJL2 because it is a highly conserved residue in immunoglobulin light chains sequence alignments (*Williams et al.*, *1996*). Opposite to this neutral mutation, Tyr36Phe did not affect the structure suggesting that the hydroxyl group of the tyrosine is not playing a relevant role (RSA 0.6%, $\Delta\Delta G = -0.04$ kcal mol$^{-1}$). While Arg24His (RSA 10%, $\Delta\Delta G = -2.70$ kcal mol$^{-1}$) performed in this work was more destabilizing than reported mutation Arg24Gly ($\Delta\Delta G = -1.52$ kcal mol$^{-1}$) (*Del Pozo-Yauner et al.*, *2008*), other effects should be considered like Phe2 reorientation to the upper hydrophobic core to compensate the absence of the Arg24 guanidinium group (*Del Pozo-Yauner et al.*, *2014*). Interestingly, Phe2 is not among the residues with higher SDRIs in 6aJL2 pointing out that compensatory effects might be attributable to the flexibility of the lateral chain.

This prompted us to assess about the role of residues or protein segments with the lowest SDRIs. Since proteins can interact with other proteins and other macromolecular, important residues not only maintain the connectivity along the tertiary structure but also maintaining quaternary structure as seen in the complex of the estrogen receptor with DNA by modifying the affinity (*Deegan et al.*, *2010*). We found that the most flexible and unconnected regions were associated with active site functions, as in the case of apoflavodoxin, to which the cofactor, flavin mononucleotide, binds (*Genzor et al.*, *1996*). In chymotrypsin inhibitor, the larger loop, which is flexible and unconnected, harbors the active site (*Jackson et al.*, *1993*). However, not all mutations can enhance the stability while preserving the original function of a protein. In the case of T4 lysozyme, some mutants were found to be more stable but resulted in losses of the protein's original function (*Shoichet et al.*, *1995*). If the purpose is to modify the function of a protein, potential mutations should be assessed by other means, such as evolutionary multiple sequence alignment (*Alexander et al.*, *2009*; *Halabi et al.*, *2009*). If the aim is only to increase the protein stability, a good approach could be locating the less structurally important residues and generating changes that benefit the formation of salt bridges as shown in apoflavodoxin. Mutations localized on the surface and designed to establish salt bridges were able to increase the overall stability in apoflavodoxin (*Campos et al.*, *2004b*). Remarkably, these mutations were performed on the low-scored SDRI residues.

Since folding requires certain flexibility degree, we analyzed the relationship between kinetic data with SDRIs. Other experiments performed in chymotrypsin inhibitor were folding/unfolding kinetics (Fig. 7) (*Itzhaki, Otzen & Fersht*, *1995*). In this case, the SDRIs showed a better correlation with the $\Delta\Delta G$ unfolding kinetic values (75% from the PCA analysis) than the $\Delta\Delta G$ folding kinetic values (57% from the PCA analysis). We would like to reiterate that folding is a dynamic rearrangement of the network because longer times and other conditions are required for better simulations of the protein folding/unfolding pathways.

Our last validation was performed over an intrinsically disordered peptide, resolved by NMR in solution, which only showed a small structured portion of the peptide (*Chandrashekaran et al.*, *2015*). Before applying the network strategy described here, the RMSD of each one of the 20 frames was calculated by comparing them against the first frame (Fig. 8A). It was evident that there is a remarkable structural motion freedom at the N-terminus, even higher than those obtained by molecular dynamics of the other proteins described here (Figs. 2, 4 and 8B). In order to follow the methodology proposed here, RMSF values of the main chain of each residue were calculated using RCI server (*Berjanskii & Wishart*, *2013*). An important difference compared to globular proteins was the distribution statistics of the SDRIs, having the highest value dispersion dissimilar to a normally distributed population (Table 1 and Fig. 4H). The most connected and less flexible residues were located in the hairpin, shown in Fig. 8C, which contributes to the scaffolding functionality allowing phosphorylation of Ser211. Also, the plasticity of the disordered N-terminus would enable this peptide to bind multiple components of the signaling pathway in which it is involved (*Chandrashekaran et al.*, *2015*).

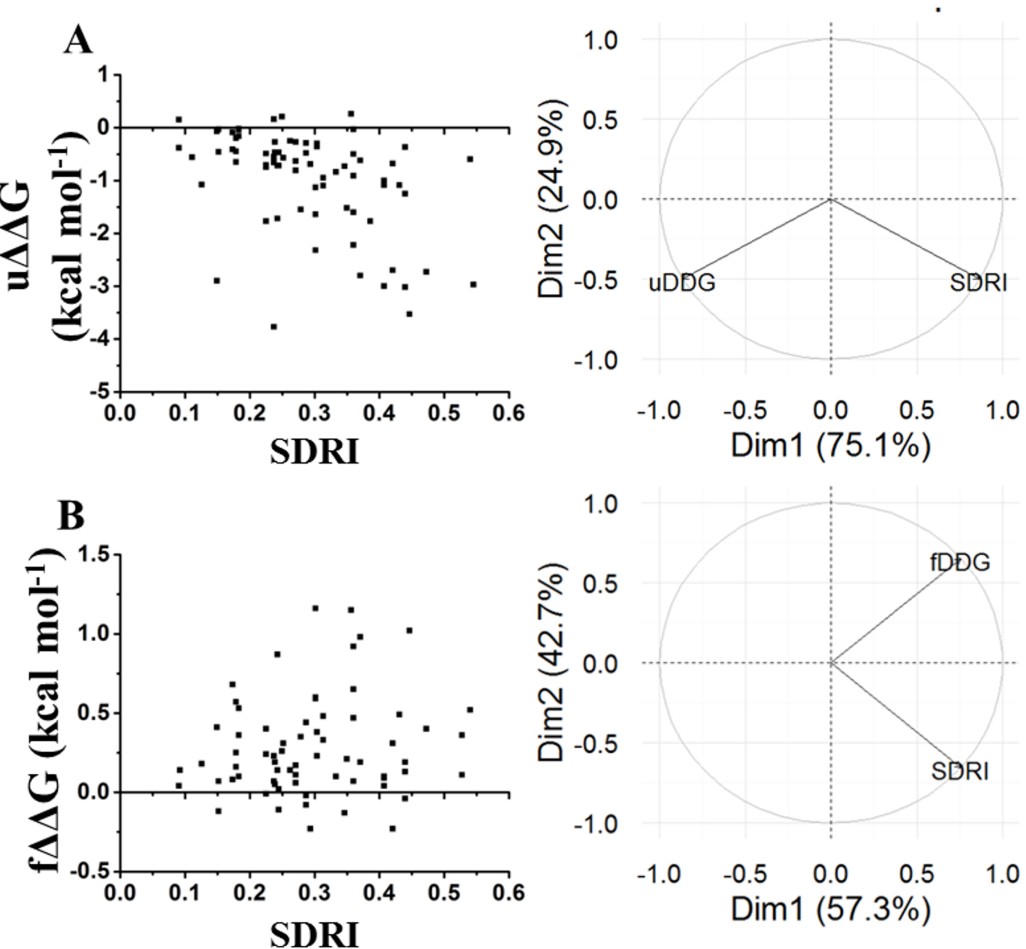

**Figure 7  Comparison of SDRIs with the kinetic unfolding and kinetic refolding processes of the chymotrypsin inhibitor.** Unfolding kinetics exhibited a better correlation with SDRIs as shown in circular biplots. DIM1 refers to SDRIs ($H_{i*}\mathrm{RMSF}_i^{-1}$), and DIM2 refers to the experimental values, $\Delta\Delta G$ (DDG) or KD percent difference (see File S3). (A) $\Delta\Delta G$ Unfolding kinetics, uDDG in circular biplot; (B) $\Delta\Delta G$ Refolding kinetics, fDDG in circular biplot.

It should be noticed that applying this methodology on proteins with high structural motion might not provide enough information to predict which residues will interact with ligands. Unstructured peptides can be analyzed by other means (*Kosol et al.*, *2013*; *Shaw et al.*, *2010*). We simulated the unbound estrogen receptor and, despite the drastic change on the ranking position and the SDRIs, it is not evident which segments of the estrogen receptor dimer will recognize precise DNA sequence (see File S4). Furthermore, we decided to extend 6aJL2 simulation time to 50 ns and we only detected subtle, but not significant, changes on SDRIs for this globular protein (see File S4). Thus, appropriate molecular dynamics simulation accomplished on globular proteins is an essential step for this methodology. Moreover, our results agreed with sequence-based predictors that look up for the intrinsically disordered segments regardless protein complexity. SDRI values displayed a versatile mathematical parameter since function (1-SDRI) might be applicable to highlight disordered segments. Such disordered segments may increase the

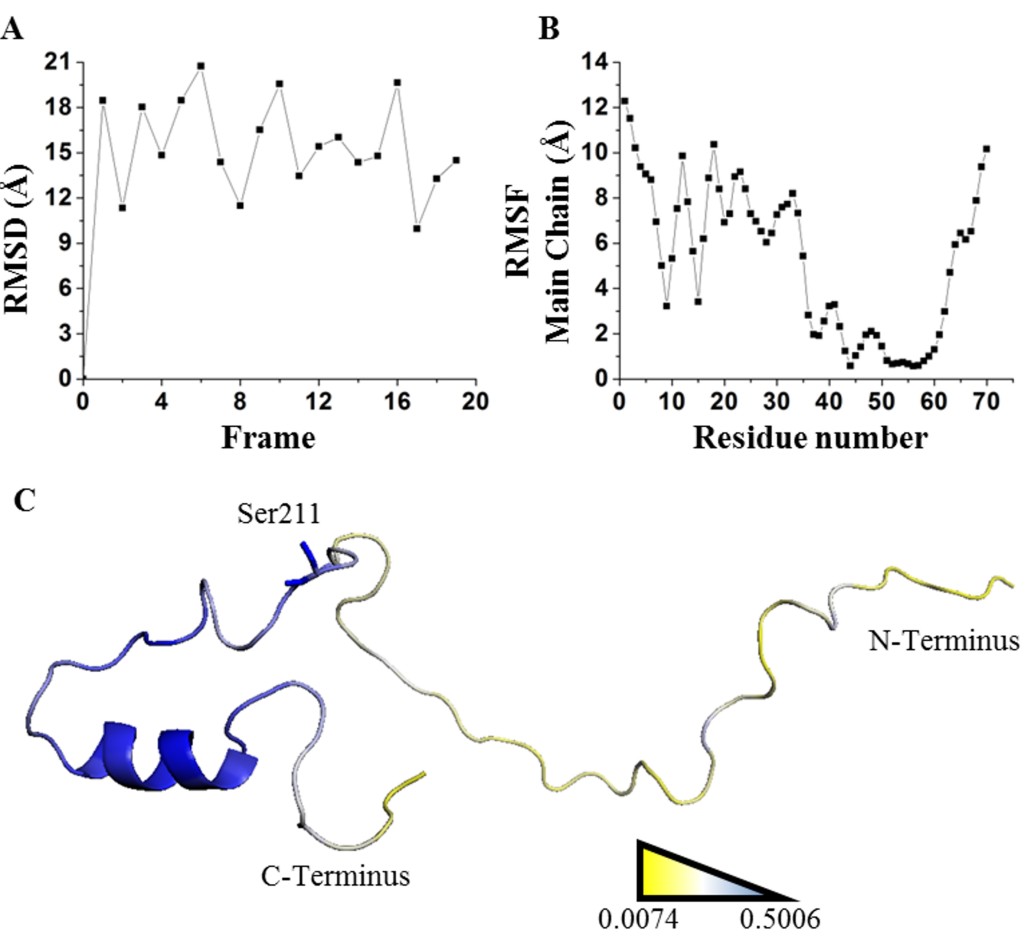

**Figure 8 Applying SDRI approach over an unstructured peptide.** Largely disordered N-terminus of suppressor of cytokine signaling 5 mammalian suppressor of Janus Kinase interaction region. (A) RMSD values by comparing each frame against first frame from PDB 2N34. (B) RMSF values of each residue showing that hairpin domain is more rigid than the rest of the protein. (C) Ranking heat map where the highest SDRI is shown in blue and the lowest SDRI is represented in yellow (see File S2).

capability of organisms to tolerate environmental challenges by diversifying the properties of their proteins to recognize several molecular partners such as cofactors, DNA, or other proteins. Higher SDRI values imply higher probabilities to modify protein stability, but there is a higher tendency to destabilization. As reported by *Tokuriki & Tawfik* (*2009*), one of the evolutionary implications of protein destabilization is that other functions or adaptations may be achieved. Finally, under the scenario of analyzing uncommon foldings or even structures bearing non-natural amino acids, our method might be suitable to assess structure stability since it does not depend on previous information such as an evolutionary multiple sequence alignment. In conclusion, we have validated a method for the analysis of globular proteins by ranking each one of their residues according to their structural relevance from a theoretical score- SDRI.

## ACKNOWLEDGEMENTS

We thank J Osuna-Quintero, JL Martínez-Morales, and M Aldana-González for the valuable discussions and advice, C Torres-Duarte for critically reviewing the manuscript, the Oligonucleotide Synthesis Unit of the Institute of Biotechnology, T Olamendi-Portugal for the DNA sequencing, and G Corzo-Burguete for facilitating the use of the computer hardware that was used for the molecular dynamics simulations

### Funding

ODLM and MIVV were supported by CONACYT 177224 and 185179, respectively. This work was supported by grants from DGAPA IN 217510. The funders had no role in study design, data collection and analysis, decision to publish, or preparation of the manuscript.

### Grant Disclosures

The following grant information was disclosed by the authors:
CONACYT: 177224, 185179.
DGAPA IN: 217510.

### Competing Interests

Vladimir N. Uversky is an Academic Editor for PeerJ.

### Author Contributions

- Oscar D. Luna-Martínez conceived and designed the experiments, performed the experiments, analyzed the data, wrote the paper, prepared figures and/or tables, reviewed drafts of the paper.
- Abraham Vidal-Limón performed the experiments, analyzed the data, contributed reagents/materials/analysis tools, reviewed drafts of the paper.
- Miryam I. Villalba-Velázquez performed the experiments, analyzed the data, reviewed drafts of the paper.
- Rosalba Sánchez-Alcalá performed the experiments.
- Ramón Garduño-Juárez analyzed the data, contributed reagents/materials/analysis tools, reviewed drafts of the paper.
- Vladimir N. Uversky analyzed the data, wrote the paper, prepared figures and/or tables, reviewed drafts of the paper.
- Baltazar Becerril analyzed the data, wrote the paper, reviewed drafts of the paper.

### Data Availability

    The raw data has been supplied as Supplemental Information.

### Supplemental Information

Supplemental information for this article can be found online at http://dx.doi.org/10.7717/peerj.2136#supplemental-information.

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
