# Peer review of "Simple approach for ranking structure determining residues"

_PeerJ, doi:10.7717/peerj.2136_

## Round 0.1 · original submission · Major Revisions

Dear Oscar ,

Thank you for submitting your manuscript for publication in PeerJ. It has been examined by three expert reviewers who have concluded that the work is potentially suitable for publication; however, it appears that all of the reviewers agree that your paper needs a major revision, and I agree with them that especially the validity of your approach is not sufficiently demonstrated. I have to take these doubts very serious and you should carefully follow the recommendations of all three reviewers (please note that one of the reviewers attached his remarks in a separated file named “annotated manuscript”). Although you should address all remarks raised by the reviewers, I would like to emphasize, apart from the validity issue already mentioned above, the remarks with respect to the methodology, especially about the length of the simulations (reviewer 1) or the replica number (reviewer 2) or statistical evaluation (reviewer 3). When preparing a revised version of the ms, be also be aware of the more enhanced introduction with a greater literature overview demanded by reviewer 2 to better support your claims. Also the request to make the code for calculation of ranking scores public in accordance to PeerJ Data Sharing policy should be fullfilled in a revised version.

I am sorry that I cannot be more positive this time and I hope that you consider the reviews useful in improving the manuscript.

Reviewer 1 ·

Basic reporting

No Comments

Experimental design

No Comments

Validity of the findings

No Comments

Annotated reviews are not available for download in order to protect the identity of reviewers who chose to remain anonymous.

Reviewer 2 ·

Basic reporting

Results and discussions are not provided as separate sections, though not sure if this is relevant. Frequently, cited literature seems to be relatively old, which in some cases seems to be insufficient to support the claims put forward by authors, e.g.:

lines 144-148: most studies related to effect of mutation on protein stability cited here are more than 20 years old, which does not seems representative at all.


Additionally, there is a number of errors that need to be corrected, some of them even hinder the understanding of the manuscript, e.g.:

line 192: Complex Receptor DNA, showed lesser residues with low theoretical values
Did you mean lower number of residues?
line 197: scores obtained by the elaborated in our study structure determining residue identifier (SDRI) - if I got it right, this should read: scores obtained by the structure determining residue identifier (SDRI) elaborated in our study
line 205: there is reference to Figure X, I guess it should have been Figure 2?
line 218: This are important observations >> These are important observations

Generally, I would also suggest that language corrections are made by a native speaker.

Experimental design

The number of replicated MD simulations are not clearly stated and corresponding standard errors are not provided. If only a single simulation was performed for each system, which is relay on the edge of proper practice, at least results of some convergence analysis need to be included.

The exact number and distributions of frames used in the analysis is not clearly specified i.e.: lines 84-87 state that 25 ns of a total simulation time was performed, extracting every 2.5 ns, which means 10 snapshots used in total. On the contrary, lines 89, 100 and 168 declare that 100 snapshots were analyzed, which were extracted every 2.5 ps. If this second sampling step was in fact used, it would sum up to only 250 ps out of 25 ns, so most of the simulation would not be covered. Such discrepancies need to be corrected throughout the text.

Validity of the findings

The developed protocols/metrics are compared with two tools that predicts protein disorder/flexibility, which is fine, but no comparison is provided with tools focused on detection of residues important for stability. This is significant drawback of the presented manuscript. In my opinion, the reported approach cannot be properly evaluated without such comparison.

Several claims need to be better explained/supported in the manuscript: e.g.

lines 190-191: it is not obvious what authors mean by the fact that the scores are well distributed through all the structure? What was the metric used for such conclusion?

lines 218-221: the statistics supporting claims here are not provided, and it is not really obvious just from Figure 2.

·

Basic reporting

The article "Simple approach for ranking structure determining residues" by Luna-Martinéz et al. tries to demonstrate the idea that it is possible to identify essential residues for the stability of a given protein by their ranking according to their connectivity and movement restrictions.

This idea is compelling, but unfortunately, authors do not sufficiently demonstrate its validity as I will describe in detail in following sections.

The article is written in well-readable English meeting professional standards especially in the introduction and background parts. However, the introduction is quite short by itself and deeper background in prediction of structure determining residues would be appreciated. Surely more than eight papers and more previous approaches (at least one or two) to the problem can be cited - surely sequence conservation approaches can be enlightening as well.

Figures are appropriately labeled, but some minor adjustments will be mentioned in the latest section of the review.

Raw data for results are made available, but the code for calculation of ranking scores should be made public as well in accordance to PeerJ Data Sharing policy.

Experimental design

Submission clearly define the research question, which is relevant and meaningful - to find structure determining residues, but as stated in previous section, submission lacks proper introduction.

Selection of testing proteins and simulation protocol is appropriate.

Description of the scoring should be however clarified in methodology section and the same definition for scoring should be used thorough the manuscript.

More information about used statistical evaluation should be also added to the methodology section.

Validity of the findings

While data presented in sequence manner in Figures 1 and 2 show nicely that proposed ranking with score is somehow connected to disorder propensity, further results about identification of structure determining residues are not robust nor statistically sound.

The plots in Figure 3 comparing stability changes and ranking values unfortunately show low correlation and while PCA analysis might show that most of variability is covered by one variable, it certainly do not explain which variable it is and how it is connected to ranking score. Also no proof is given whether PC1 is composed the same between individual systems.

As such the ranking by score lacks predictivity (no blind test of ranking to identify important residues was given) and it unfortunately lacks also usefulness to fulfill the goal to rank structure determining residues.

For this reason, the conclusions are speculative at best and manuscript has to be improved greatly in these parts. For further detailed review see next section.

Additional comments

Here I list all misconceptions or probably misinterpretation by their first mention, which would benefit from further clarification:

line 74-76 - 58.5x59x91 A^3 seems not to be cubic simulation box for 1HCQ
line 81-83 - why are two steps in simulated annealing at 400 K reported individually, when seemingly nothing changed?
line 92 and other mentions of van der Waals radius of the node - nowhere is explained how is van der Waals radius of the node defined - it is only vdW radius of central atom or are vdW radius somehow defined for full residue (amino acid or nucleotide base)?
line 100-103 - more explanation whether normalization by row do not skew the overall normalization of whole matrix is needed - especially, whether the resulting normalized matrix is still symmetrical as was former non-normalized one.
line 104 - vector and matrix notation should be used in the equation.
line 109 - here should be added definition of the score and its normalization, as later in the text it is defined several times and always differently named (lines 188 and Figure 1 - normalized theoretical score; line 197 - SDRI without further explanation; line 207 and Figure 2 1-H*RMSF^-1; ranking in Figure 3 and H*RMSF^-1 in later lines) - one definition and notation would largely enhance readability of the manuscript.
lines 137 and 140 - m in Tm and Cm should be shown in small index
line 174 - when are two atoms closer than their van der Waals radii, then their non covalent interactions can be in strongly repulsive region not only in strongly attractive one. But here I am referring to atomistic van der Waals radii, which are not exactly explained whether they are used in the node vdW definition.
line 180 - inverse distance calculations can be highly sensitive to variations within short distances, how these changes are accounted for in the current approach?
line 187 - if RMSF is really directly correlated to dynamic entropic value, then their numerical division should bring a constant value with noise - in that case, why to use RMSF to normalize dynamic entropic value?
line 192 - less residues
line 196-199 - the sentence is overcomplicated and hard to read
line 213 - probably "less flexible protein parts"?
line 217 - this statement should be enlarged with details, what can be learned from the disagreement of those predictions
line 229 - Figure 3 - how can be explained the fact that higher ranking residues usually show lower variability of experimental data or even increase in the stability, whereas low ranked residues (aka less important) show larger destabilization? This is in direct contradiction to the explanation of scoring values as presented in the text!!!
line 234 - PCA analysis allows also further decomposition of identified principal axes - it would be interesting to decompose whether first principal axis show similarities between individual proteins.
line 240-242 - how are these findings about lethal mutations represented in the data? DGG values for such events would not be probably measured, but these datapoints should be present in the evaluation of the score.
line 243-253 - where can be found visual comparison between those two similar proteins? I haven't found any data to look at.
line 251-253 - are those most stabilizing residues also the highest ranking ones? Or they are not as much connected to other residues as they are at the surface of the protein?
line 259 - why were selected residues with low RSA, when line 237 states that ranked importance of residue is irrespective to its solvent accessibility (RSA)
line 260-263 - I would expect values of score as well...
line 296-297 - Why to speculate, that current method would not work on unstructured polypeptides, when it was not tested on any of them?

---

## Round 0.2 · Major Revisions

Dear Oscar,

As the academic editor I well recognize the huge effort you made in revising the manuscript, and I fully agree with reviewer 3, Karel Berka, that the quality of manuscript was significantly improved especially in the introduction and methodological parts. But as you can see from the reviews, both reviewers that re-reviewed the ms have still a few requests that you should address prior to publication. Especially Karel Berka feels that especially the graphical representation of the data still requires quite a lot clarifications ( and also the request of reviewer 1 is in this direction).
Additionally to improving the graphical representation you should not forget that also some issues are left unanswered from reviewer 3's previous review, and as Karel Berka points out, the Perl code for calculation of SDRI should be made public in accordance with PeerJ Data Sharing policy. I hope you find the recent comments useful for revising the manuscript and that we will receive your revised version soon.

Reviewer 1 ·

Basic reporting

The work here has the potential to publish however the authors are requested to address the following issue:
1- Insert letter alphabet/protein name or PDB ID for each RMSD curve.

Experimental design

No comments

Validity of the findings

No comments

·

Basic reporting

The quality of manuscript was significantly improved especially in the introduction and methodological parts. But there are still some issues left unanswered from my previous review and graphical representation of the data still requires quite a lot clarifications.

Experimental design

no further points

Validity of the findings

• lines 335-351 - Mutations of 6aJL2 should be reported in consecutive order to show whether method really work:
mutation Q6N Y36F R24H R24G I29G
SDRI 0.5055 0.3274 0.3273 0.3273 0.2057
ddG -3.34 -0.04 -2.70 -1.52 -3.61
and from this side-by side comparison, it seems, that SDRI values do not show any predictivity about the destabilization effect. (low SDRI should be connected with largest potential to stabilize protein, which is clearly not the case within presented data)
• line 716 - Figure 5 – From the figure it is clear, that PCA mapping is different for each system, and actually it strengthen my former argument, that whole SDRI analysis is not predictive enough for the protein destabilization, because it shows (similarly as the scatter plots in the same figure) that there is really a small consensus between experimental data and SDRIs.
• lines 648-655 and 719 - Figure 6 caption says that blue color belongs to the highly connected nodes with highest SDRI, but Figure 6 insets show that blue colored residues have values around 1. So it seems that inset is not referring to SDRI itself but to SDRI ranking order, which is counterintuitive. Please do select just ONE representation of SDRI values thorough the manuscript and do NOT change it between individual figures and text. I have counted at least three different ways of presentation of the same SDRI data (SDRI rank, normalized SDRI, 1-SDRI) which is still highly confusing even though it was even more confusing in the previous version.
• Last important point is that while raw data for results are made available, the Perl code for calculation of SDRI should be made public as well in accordance to PeerJ Data Sharing policy.

Additional comments

• line 708 - Figure 2 – according to RMSD, protein D (at least, I guess, there are no letters present, even thought ) is not stable at all (RMSD of backbone over 5) and it should be avoided in further analyses
• Consequently - all Figures should include self-readable description within Figure itself, not only in the caption (which is upon submission several pages away from Figure itself)
• line 716 - Figure 5 – PCA mapping analysis is unreadable until at 200 % magnification
• Comparison between thermophilic and hyperthermophilic Cold shock proteins should be taken side by side for easy comparison - in most cases, their position in Figures are not aligned.

---

## Round 0.3 · accepted · Accept

I feel that your last revision met most of the reviewers' requests and that this version is now acceptable for publication. However, in the production process you need still to improve figure 5. I find the new revised version even more difficult to read when printed on A4 or letter size paper, the numbers and letters on the variables factor map are smaller than 8pt which is unacceptable for publication. I think you simply can't pack so many panels on one figure and I would split the figure into two. You probably need to discuss this with the PeerJ production staff.